# Phytochemical characterization, anticancer potential, and nanoemulsion-based delivery of *Chiliadenus montanus*

**Nour Aboalhaija**[1]*, **Hala Abulawi**[1], **Rania Hamed**[1], **Mohammad Alwahsh**[1], **Fatma Afifi**[2], **Heba Syaj**[1], **Elham Abusharieh**[1], **Ismail Abaza**[2]

**1** Department of Pharmaceutical Sciences, Faculty of Pharmacy, Al-Zaytoonah University of Jordan, Amman, Jordan, **2** Department of Pharmaceutical Sciences, School of Pharmacy, The University of Jordan, Amman, Jordan

* n.aboalhaija@zuj.edu.jo

## Abstract

*Chiliadenus montanus* (Vahl) Boiss. (Asteraceae) is a pharmacologically significant plant with different potent pharmacological properties. This study aimed to evaluate the phytochemical and anticancer activity of *C. montanus*, and to develop nanoemulsions (NEs) to enhance pulmonary delivery for lung carcinoma treatment. For that ethanol and water extracts, along with petroleum ether, chloroform, ethyl acetate, and methanol fractions, were assessed for total phenol and flavonoid contents, antioxidant activity, and cytotoxicity against H1299 and A549 lung cancer cell lines. The results showed that ethyl acetate fraction exhibited the highest phenol ($47.94 \pm 0.32$ mg GAE/g of DW) and flavonoid ($20.34 \pm 1.48$ mg rutin/g of DW) contents, while the ethanol extract showed the most potent antioxidant activity ($IC_{50} = 322.1$ µg/mL) and selective cytotoxicity ($IC_{50} = 641.2$ µg/mL) against H1299 cells. Nevadensin, chlorogenic acid, and sorbifolin were identified as the major constituents of the ethanol extract using liquid chromatography-mass spectrometry (LC-MS) analysis. Gas chromatography-mass spectrometry (GC-MS) analysis revealed α-phellandrene, 1,8-cineole, and α-cadinol as the lead volatile constituents. The major volatile compound of the aroma profile of the aerial parts, determined by solid phase micro extraction (SPME) was 1,8-cineole. Spontaneous emulsification was used to formulate ethanolic NE (S1-S4 NEs) with varying concentrations of ethanol extract, surfactant, cosurfactant, and oil phase. The optimal S4 NE demonstrated thermodynamic stability, appropriate pulmonary pH, and droplet sizes below 100 nm. These findings highlight the promising potential of *C. montanus* NE as a stable pulmonary drug delivery system for lung carcinoma therapy.

**Data availability statement:** All data generated or analyzed during this study are included in this published article.

**Funding:** This research was funded by Deanship of Scientific Research and Innovation at Al-Zaytoonah University of Jordan (2022-2023/07/23). The funders had no role in study design, data collection and analysis, decision to publish, or preparation of the manuscript.

**Competing interests:** The authors declare no conflicts of interest. The funders had no role in the design of the study; in the collection, analyses, or interpretation of data; in the writing of the manuscript; or in the decision to publish the results.

**Abbreviations:** DLS, Dynamic Light Scattering; DW, Dry weight; GC-MS, Gas Chromatography-Mass Spectrometry; HD, Hydrodistillation; LC-MS, Liquid Chromatography-Mass Spectrometry; MDS, Mean Droplet Size; NEs, Nanoemulsions; PDI, PolyDispersity Index; SPME, Solid-Phase Micro-Extraction; TEM, Transmission Electron Microscopy; TFC, Total Flavonoid Content; TPC, Total Phenol Content; ZP, Zeta Potential.

## 1. Introduction

The genus *Chiliadenus* (Asteraceae, formerly Compositae) includes three species native in Jordan: *C. montanus* (Vahl) Brullo (syn. *Varthemia montana* Vahl (Boiss.)), *C. candicans* (Delile) Brullo (syn. *V. candicans* Delile (Boiss.)), and *C. iphionoides* (Boiss. & Blanche) Brullo (syn. *V. iphionoides* Boiss. & Blanche) [1]. Among these, *C. montanus*, commonly known as desert Varthemia, is a perennial bushy herb characterized by small leaves, numerous branches, and aromatic, hairy, and sticky stems, reaching a height of 20–50 cm [2]. It thrives in rocky and semi-arid regions, blooming between September and December with distinctive tubular yellow flowers. Various species formerly assigned to *Varthemia* have demonstrated significant pharmacological activities, including antibacterial, antifungal, antioxidant, anti-inflammatory, anticancer, antispasmodic, and antiplatelet effects [2–5].

Phytochemically, *C. montanus* is known to be rich in volatile oils, both volatile and non-volatile terpenoids and flavonoids, -mainly methoxylated-. All these secondary metabolites justify the presence of a wide range of biological activities, including antibacterial, antifungal, antioxidant, anti-inflammatory, anticancer, antispasmodic, antiplatelet, and hypoglycemic effects [2,6–12].

The species' possible benefits in the treatment of Alzheimer's disease are suggested by El-Bsoumy et al. [10]. Several other researchers have emphasized different traditional uses of *C. montanus,* including its benefits in management of diarrhea, kidney problems, stomachaches, and chest ailments, especially recommended in Mediterranean countries [2,12,13]. Keeping in mind that environmental factors can significantly influence the phytochemical profiles of medicinal plants, further investigations of their extracts are needed in order to determine the effect of these factors in the biological activity.

This present study aimed to evaluate the chemical composition and biological activities of extracts and fractions of *C. montanus* harvested in Jordan. Their total phenol and flavonoid contents were analyzed, along with their antioxidant and antiproliferative activities. To enhance and expand its phytochemical and biological activities, pharmaceutical formulation of its ethanol extract was approached. The ethanol extract of *C. montanus* was incorporated into a nanoemulsion (NE) formulation and assessed as a potential drug delivery system for lung cancer therapy. By investigating its cytotoxic potential and suitability for pulmonary delivery, this research highlights the therapeutic value of *C. montanus* and its potential role as adjunctive phytotherapy for conventional chemotherapy. The composition of the active extract was evaluated using LC-MS. Additionally, the volatile oil composition of the hydro-distilled essential oil and aroma profile of the aerial parts of this species were determined to obtain a complete picture of its phytochemical composition.

## 2. Materials and methods

### 2.1. Plant material

*Chiliadenus montanus* was collected from the Balqa region (32.037° N, 35.7288° E) of Jordan, in late September 2022. The plant was authenticated by one of the authors

using descriptive references and in comparison, with the herbarium specimen in the School of Pharmacy, The University of Jordan, Amman. Voucher specimens were kept in the School of Pharmacy (FMJ-AST95–3). *C. montanus* is characterized by its multiple branches, hairy appearance, sticky stems, and bushy perennial form with small leaves and a woody base, typically ranging in length from 20 to 50 cm [14].

## 2.2. Essential oil extraction by hydrodistillation (HD) and aroma profile determination by solid-phase micro-extraction (SPME)

For the isolation of the essential oil of *C. montanus* aerial parts of the plant were subjected to hydrodistillation in a Clevenger apparatus for 3 hours. The resulting oil was pooled, dried over anhydrous sodium sulfate ($Na_2SO_4$), and stored at 4 °C in amber glass vials until analysis [15]. For aroma profiling, solid-phase micro-extraction (SPME) was employed to capture and analyze the volatile compounds as described earlier by Afifi et al. [16].

## 2.3. Gas Chromatography-mass spectrometry (GC-MS)

The volatile organic compounds emitted of *C. montanus* essential oil were analyzed using GC-MS [16]. Compound identification was performed using internal digital libraries, including those from NIST and Wiley, by comparing the acquired retention indices (RI) with published values for $C_8$-$C_{20}$ *n*-alkanes, measured under similar conditions [17].

For qualitative analysis, oil samples obtained by HD and (SPME) were diluted in GC-grade hexane and analyzed on a Varian Chrompack CP-3800 GC/MS/MS-200 (Saturn) equipped with a DP-5 capillary column (30 m × 0.25 mm i.d., 0.25 μm film). Helium was used as the carrier gas (0.9 mL/min). The oven was programmed from 60°C (1 min, isothermal) to 250°C at 3°C/min. MS conditions were: ionization voltage 70 eV and source temperature 180°C. A homologous series of n-alkanes ($C_8$–$C_{20}$) was run under identical conditions. Identification was based on spectral library matching, calculated RI values compared to literature data [17,18].

Quantitative composition (% area) was determined using a Hewlett-Packard HP-8590 GC equipped with a split–splitless injector (split ratio 1:50) and FID detector. Separation was achieved on an Optima-5 capillary column (30 m × 0.25 mm i.d., 0.25 μm film) with oven temperature programmed from 60°C to 250°C at 10°C/min, then held for 5 min. Relative peak areas were used to calculate compound concentrations [18]. The experimental RI values were calculated using the following formula 1 below:

$$RI = 100 \left( n + (N - n) \times \frac{(Logtr\ (unknown) - logtr\ (n))}{logtr\ (N) - logtr\ (n)} \right)$$

(1)

where RI represents the retention time measured from the time of the unretained small molecule, and *n* and *N* represent the numbers of carbon atoms in the smaller and larger alkanes, respectively. A Hewlett-Packard HP-8590 gas chromatograph equipped with a split-spitless injector (split ratio 1:50) and a flame ionization detector (FID) was utilized for quantitative analysis (% area). The concentration of the identified compounds was determined by measuring the relative peak areas of the oil components.

## 2.4. Extracts preparation

The ethanol extract of *C. montanus* was prepared by soaking the dried and crushed aerial parts in 70% ethanol for three weeks at room temperature (RT). Following this period, ethanol was evaporated under vacuum using a rotary evaporator set at 40 °C. The crude extract was first subjected to defatting by liquid-liquid extraction using petroleum ether in a separatory funnel to remove chlorophyll and fatty materials. This process was repeated until the petroleum ether layer became colorless, ensuring the complete removal of non-polar impurities. The defatted water methanol layers were collected and subsequently fractionated based on polarity using a sequential liquid-liquid extraction approach. Chloroform was

employed to isolate non-polar constituents, followed by ethyl acetate to extract intermediately polar phytochemicals. The remaining water methanol layer was designated as the methanol fraction, containing the highly polar components. Additionally, water and ethanol extracts were prepared by soaking the plant material in water and 70% ethanol, respectively, without further fractionation.

## 2.5. Determination of total phenol (TPC) and total flavonoid contents (TFC)

The TPC of *C. montanus* extracts was determined using the Folin-Ciocalteu method. Gallic acid in methanol served as a standard, and five serial dilutions (100, 50, 25, 12.5, and 6.125 µg/mL) were prepared. Absorbance readings were taken at 765 nm in triplicate, with methanol used as a blank. The total phenol content was calculated using the generated equation and expressed as gallic acid equivalents (GAE) [19].

To determine TFC, aluminum chloride was utilized. After incubation for 30 min at RT, the absorbance was measured at 415 nm. Rutin served as the calibration standard, with five concentrations (100, 50, 25, 12.5, and 6.125 µg/mL) prepared in methanol. Experiments were carried out in triplicate, and the results were calculated using the generated equation and averaged [19].

## 2.6. Antioxidant activity

To prepare the DPPH reagent, approximately 4 mg of DPPH powder was dissolved in 100 mL of methanol in an Erlenmeyer flask. The mixture was agitated using a magnetic stirrer at 60–100 rpm for 5 min to ensure complete dissolution. To protect the DPPH solution from light and evaporation, the flask was wrapped in foil and sealed. Ascorbic acid was used as a standard to establish a calibration curve. Ten different concentrations were prepared (1000, 500, 250, 125, 62.5, 31.25, 15.625, 7.815, 3.91, and 1.955 µg/mL). In a test tube, 3 mL of the DPPH solution was mixed with 2 mL of each extract. The reaction was allowed to proceed for 30 min at RT in complete darkness.

After the incubation period, the absorbance of each extract was measured at 517 nm. The percentage of inhibition was calculated using the formula 2 below:

$$I\% = (A° - Ax) / A° \times 100 \tag{2}$$

where A° represents the absorbance of the control (DPPH solution without extract), and Ax is the absorbance of the extract at a specific concentration. The $IC_{50}$ value was determined as the concentration required to inhibit 50% of the DPPH radicals.

## 2.7. In vitro cytotoxicity on cancer cell lines

Human lung cancer cell lines A549, H1299, and normal fibroblasts were obtained from the American Type Culture Collection (ATCC, Manassas, VA, USA). A549 (CCL-185) and H1299 (CRL-5803) cells were cultured in RPMI-1640 medium, while fibroblasts (PCS-201–018) were cultured in Dulbecco's Modified Eagle's Medium (DMEM). Both media were supplemented with 10% fetal bovine serum (Cytevia, Austria), 1% L-glutamine, penicillin-streptomycin, and HEPES buffer (EuroClone S.P.A., Italy). Cells were incubated at 37 °C in a humidified 5% $CO_2$ atmosphere and subcultured every 2–3 days, with passages limited to 20 [19]. Cells were seeded into 96-well plates (10,000 cells/100 µL/well) (SPL Life Sciences Co., Korea). After reaching 70% confluence, cells were treated with serially diluted *C. montanus* extract (2000, 1000, 500, 250, and 125 µM) in five replicates. After 48 hours of incubation, the media were replaced with fresh media containing 15 µL (0.5 mg/mL in each well) of MTT (3-[4,5-dimethyl-thiazol-2-yl]-2,5-diphenyl tetrazolium bromide; ATCC® 30–1010 K) and incubated for 3 hours. Formazan crystals were dissolved with solubilization solution, and the absorbance was measured at 570 nm using an HTX multi-mode microplate reader (BioTek, USA). Data were analyzed using GraphPad Prism 8.0.2.

## 2.8. LC-MS analysis

LC-MS analysis was performed on a Bruker Daltonik Impact II ESI-Q-TOF mass spectrometer (Bremen, Germany) coupled to a Bruker Elute UPLC system equipped with an Apollo II ion funnel electrospray source. Stock solutions of reference standards were prepared in dimethyl sulfoxide (DMSO) and diluted with acetonitrile to determine exact mass-to-charge ratios (m/z) and retention times. All solvents and reagents, including acetonitrile, methanol, water, and formic acid, were LC/MS grade. Samples were dissolved in 2 mL DMSO, diluted to 50 mL with acetonitrile, centrifuged at 4000 rpm for 2 min, and 1 mL of the supernatant was transferred to an autosampler vial; 3 μL was injected for analysis. The instrument was operated with a capillary voltage of 2500 V, nebulizer gas pressure of 2.0 bar, dry nitrogen flow of 8 L/min at 200 °C, and an ion source block temperature of 180 °C. Mass accuracy was < 1 ppm, resolution 50,000 FSR (Full Sensitivity Resolution), and TOF repetition rate up to 20 kHz. Chromatographic separation was achieved on a Bruker Solo 2.0 C18 UHPLC column (100 mm × 2.1 mm, 2.0 μm) maintained at 40 °C with a flow rate of 0.51 mL/min. The mobile phase consisted of (A) water with 0.05% formic acid and (B) acetonitrile, using the following gradient: 0–27 min, linear increase from 5% to 80% B; 27–29 min, 95% B; 29.1 min, return to 5% B. The total run time was 35 min, and analyses were performed in both positive and negative ionization modes. Compounds were identified by comparing high-resolution m/z values and retention times with those of authentic standards analyzed under identical conditions [20].

## 2.9. Preparation of C. montanus Nanoemulsions (NEs)

The ethanol extract was selected for NE formulation to enhance pulmonary delivery based on the high TPC and TFC of this extract as well as its strong antioxidant and antiproliferative activities. This extract was incorporated into an oil-in-water (O/W) NE using the spontaneous emulsification method [21]. Briefly, 100 mg of the extract was dissolved in 1 mL of ethanol, and volumes of either 250 μL or 500 μL of this solution were mixed with 2.25 mL or 1.75 mL of ethanol under continuous stirring for 6 minutes at RT. Oleic acid and Tween 20 were then added to the mixture, followed by the dropwise addition of deionized water to facilitate the formation of the NE. The detailed composition of the prepared formulations (S1-S4 NEs) is provided in Table 1.

The resulting NEs were sonicated for 30 minutes to ensure transparency and remove any bubbles. The formulations were stored at room temperature (RT) or 4 °C for further stability evaluation. This method allowed for the preparation of stable NEs with varying compositions, which were subsequently characterized for their physicochemical properties and stability. The S4 NE was subjected to thermodynamic stability studies [22].

## 2.10. Characterization of C. montanus NEs

### 2.10.1. Thermodynamic stability.
Thermodynamic stability of *C. montanus* NEs was assessed as described by Hamed et al. [21] using centrifugation (Hermle centrifuge Z216MK, Germany) at 3500 rpm for 15 min, and heating-cooling cycles (six cycles at 45 °C and 4 °C over 48 hours). Stability was further evaluated using three freezing-thawing cycles (−21 °C to 25 °C) over 48 hours.

**Table 1. The composition of *C. montanus* NEs prepared using the spontaneous emulsification method.**

| Formulations | Ethanol Extract (μg) | Ethanol (mL) | Tween 20 (mL) | Oleic acid (mL) | Deionized water (mL) | Storage condition |
|---|---|---|---|---|---|---|
| S1 | 500 | 1.75 | 1.25 | 0.5 | 1.00 | RT |
| S2 | 250 | 2.25 | 1.50 | 0.5 | 0.75 | 4°C |
| S3 | 250 | 2.25 | 1.25 | 0.5 | 1.00 | RT |
| S4 | 250 | 2.25 | 1.25 | 0.25 | 1.25 | RT |

NEs: Nanoemulsions, RT: Room temperature.

**2.10.2. pH and viscosity measurements.** The pH of *C. montanus* NEs was measured using a Jenway Benchtop pH Meter (Cole-Parmer Ltd, Model 3510, UK). Additionally, the viscosity of *C. montanus* NE was determined with a rotational rheometer (QC Rheolab, Anton Paar, Austria) at 25 °C, with shear rates ranging from 10 to 1000 s$^{-1}$.

**2.10.3. Size distribution and zeta potential.** Mean droplet size (MDS), polydispersity index (PDI), and zeta potential (ZP) of *C. montanus* NEs were measured using dynamic light scattering (DLS) (Nicomp Nano Z3000, Entegris, USA) at 25 °C. Measurements were conducted in triplicate, and reported initially and weekly for two months.

**2.10.4. Transmission electron microscopy (TEM) analysis.** NE morphology was analyzed using a transmission electron microscope (TEM, FEI Morgani 268, Holland). Samples were placed on carbon-coated copper grids, dried for 30 minutes, and imaged at an accelerating voltage of 120 kV.

**2. 10.5. Physical stability.** Physical stability of *C. montanus* NEs, including phase separation, homogeneity, and color change, was monitored visually for 60 days at RT and at 4 °C.

## 3. Results

### 3.1. Gas chromatography-mass spectrometry (GC-MS) analysis of C. montanus

The hydrodistillation of C. montanus yielded a bright yellow essential oil with an intense characteristic odor. The yield was 0.094% (v/w). GC-MS analysis revealed 30 constituents, with monoterpenes as the dominant chemical class (74.6% of the total composition). Mnoterpenes were further categorized into hydrocarbon monoterpenes (37.7%) and oxygenated monoterpenes (36.9%). Additionally, sesquiterpenes hydrocarbons contributed to 14.9%, while miscellaneous compounds accounted for 7.5% of the composition. The most abundant compounds identified in the HD extract were α-phellandrene (28.60%), 1,8-cineole (23.30%), and α-cadinol (11.90%) as shown in Table 2.

Solid-phase micro-extraction (SPME) analysis revealed a higher proportion of monoterpenes (93.0%), distributed as oxygenated monoterpenes (50.4%) and hydrocarbon monoterpenes (42.6%). Sesquiterpenes were detected in significantly lower quantities (1.4%), with oxygenated sesquiterpenes comprising 0.6% and hydrocarbon sesquiterpenes 0.8%. The predominant volatile constituents of the aroma profile were 1,8-cineole (35.00%), sabinene (12.70%), and α-artemisia triene (11.40%).

### 3.2. Total phenol (TPC) and total flavonoid contents (TFC) determinations

The quantitative analysis of TPC and TFC in different solvent extracts of C. montanus is summarized in Table 3.

The ethyl acetate fraction exhibited the highest TPC (47.94±0.32 mg GAE/g), followed by ethanol extract and methanol fraction. Similarly, TFC levels were highest in ethyl acetate fraction (20.34±1.48 mg rutin/g), followed by ethanol extract and chloroform fraction. Water extract displayed the lowest TPC and TFC.

### 3.3. Antioxidant activity

The results indicated that the ethanol extract exhibited the lowest IC$_{50}$, followed by the chloroform and the petroleum ether- fractions) indicated the high antioxidant activity of ethanol extract. The water extract demonstrated the lowest antioxidant activity, with an IC$_{50}$ of 1077.7±60.9 μg/mL. The results are shown in Table 4.

### 3.4. Antiproliferative activity

The cytotoxic effects of the extracts and fractions were evaluated against human lung cancer cell lines A549, H1299, and normal fibroblasts, were obtained from the American Type Culture Collection (ATCC, Manassas, VA, USA). The chloroform and petroleum ether fractions were excluded from further studies due to their toxicity to normal fibroblast cell lines. The water extract exhibited weak cytotoxicity against the H1299 cell line, with an IC$_{50}$ of 984.4 μg/mL, but showed no activity against the A549 cell line (Table 5 and Fig 1).

**Table 2. Composition of the volatile oils of the fresh aerial parts of C. montanus obtained by HD and SPME.**

| No. | KI Lit.* | KI Exp. | Compound | HD % | SPME % |
|---|---|---|---|---|---|
| 1 | 923 | 909 | α-Artemisia triene[a] | – | 11.40 |
| 2 | 932 | 933 | α- Pinene[a] | 00.10 | 0.40 |
| 3 | 945 | 965 | α- Fenchen[a] | 00.20 | 0.40 |
| 4 | 964 | 941 | Camphene[a] | – | 10.00 |
| 5 | 969 | 1019 | Sabinene[a] | – | 12.70 |
| 6 | 974 | 1005 | β- Pinene[a] | 00.20 | 5.20 |
| 7 | 988 | 1016 | Myrcene[a] | – | 0.10 |
| 8 | 999 | 1220 | Yomogi alcohol[b] | 06.00 | 0.30 |
| 9 | 1002 | 1063 | α-Phellandrene[a] | 28.60 | 0.10 |
| 10 | 1008 | 1052 | Carene[a] | 03.00 | 0.80 |
| 11 | 1014 | 1181 | α-Terpinene[a] | 00.90 | 1.80 |
| 12 | 1020 | 1172 | β- Cymene[a] | 00.50 | 2.20 |
| 13 | 1024 | 8617 | Limonine[e] | – | 3.60 |
| 14 | 1025 | 1031 | β-Phellandrene[a] | – | 2.10 |
| 15 | 1026 | 1198 | 1.8 Cineole[b] | 23.30 | 35.00 |
| 16 | 1034 | 1225 | Santolina alcohol[b] | 00.20 | 2.60 |
| 17 | 1054 | 1058 | Terpinene <γ[a] | – | 3.20 |
| 18 | 1065 | 1252 | Sabinene hydrate[b] | 00.80 | 3.80 |
| 19 | 1086 | 1204 | Terpinolene[a] | 03.40 | – |
| 20 | 1095 | 1301 | Linalool[b] | 01.10 | 0.10 |
| 21 | 1135 | 1602 | E-Pinocarveol[b] | 00.60 | 0.60 |
| 22 | 1136 | 1304 | E-p-Menthenol[e] | 01.70 | – |
| 23 | 1141 | 999 | Camphor[b] | 00.90 | – |
| 24 | 1160 | 1378 | Pinocarvone[b] | 02.70 | 0.10 |
| 25 | 1165 | 1665 | Borneol[b] | 00.50 | – |
| 26 | 1186 | 1641 | α- Terpineol[b] | 01.20 | 0.10 |
| 27 | 1194 | 1670 | Myrtenol[b] | 00.20 | – |
| 28 | 1254 | 1412 | Bornyl acetate[e] | 00.20 | – |
| 29 | 1277 | 1641 | Ethyl chrysanthemumate[e] | 00.20 | – |
| 30 | 1288 | 1484 | Lavandulyl acetate[e] | 01.70 | – |
| 31 | 1298 | 1698 | Carvacrol[b] | 00.20 | – |
| 32 | 1311 | 1536 | Pinocarvyl acetate[e] | 00.10 | – |
| 33 | 1439 | 1403 | Aromadendrene[c] | – | 0.30 |
| 34 | 1522 | 1043 | Cadinene[c] | – | 0.30 |
| 35 | 1548 | 1653 | α-Cadinol[d] | 11.90 | 0.40 |
| 36 | 1582 | 1495 | Caryophyllene oxide[d] | 01.00 | 0.40 |
| 37 | 1644 | 1687 | α-Muurolol[d] | 00.80 | – |
| 38 | 1665 | 1705 | Intermedeol[d] | 01.20 | – |
| 39 | 1783 | 1677 | Eudesmol acetate[e] | 03.60 | – |
| | **Monoterpenes** | | | 74.6% | 93.0% |
| | Oxygenated monoterpenes[a] | | | 36.9% | 50.4% |
| | Monoterpene hydrocarbons[b] | | | 37.7% | 42.6% |
| | **Sesquiterpenes** | | | 14.9% | 1.4% |
| | Oxygenated sesquiterpenes[c] | | | 0% | 0.6% |
| | Sesquiterpene hydrocarbons[d] | | | 14.9% | 0.8% |

*(Continued)*

| No. | KI Lit.* | KI Exp. | Compound | HD % | SPME % |
|-----|----------|---------|----------|------|--------|
|  | Miscellaneous[e] |  |  | 07.5% | 3.6% |
|  | Total identified |  |  | 97.0% | 98.0% |

*Lit KI: Reported Kovats Index, [a]Oxygenated monoterpenes, [b]Monoterpene hydrocarbons, [c]Oxygenated sesquiterpenes, [d]Sesquiterpene hydrocarbons, and [e]Miscellaneous

**Table 3. Total phenol content and total flavonoid content of C. montanus extracts.**

| Extract | TPC (GAE mg of gallic acid/g) | TFC (mg of rutin/g) |
|---------|-------------------------------|----------------------|
| Ethanol* | 20.54 ± 0.41 | 10.27 ± 0.74 |
| Ethyl acetate** | 47.94 ± 0.32 | 20.34 ± 1.48 |
| Methanol** | 19.19 ± 0.27 | 5.28 ± 0.34 |
| Chloroform** | 6.55 ± 0.50 | 9.71 ± 2.03 |
| Water* | 1.03 ± 0.56 | 0.12 ± 0.06 |
| Petroleum ether** | 1.47 ± 0.34 | 1.63 ± 0.71 |

Data are shown as the average of triplicates ± SD, n = 3; *Extract; **Fraction.

**Table 4. Antioxidant activity of C. montanus extracts.**

| Extract | $IC_{50}$ μg/mL |
|---------|-----------------|
| Ethanol* | 322.1 ± 39.9 |
| Ethyl acetate** | 843.3 ± 171.7 |
| Methanol** | 889.9 ± 45.3 |
| Chloroform** | 350.3 ± 68.1 |
| Water* | 1077.7 ± 60.9 |
| Petroleum ether** | 488.4 ± 109.9 |

Data are shown as the average of triplicates ± SD, n = 3; *Extract; **Fraction.

The methanol fraction displayed no cytotoxicity against the A549 or H1299 cancer cells. Ethanol extract and ethyl acetate fraction exhibited potent cytotoxic activity against the A549 and H1299 cell lines without affecting gingival fibroblasts. The $IC_{50}$ for the ethanol extract was 476.5 and 641.2 μg/mL against the A549 and H1299 cell lines, respectively. Similarly, the $IC_{50}$ for the ethyl acetate fraction was 468.0 and 877.6 μg/mL against the A549 and H1299 cell lines, respectively.

### 3.5. Liquid chromatography-mass spectrometry (LC-MS)

The active ethanol extract and ethyl acetate fraction were subjected to LC-MS analysis to identify the phytochemical composition to link the identified compounds with their cytotoxic activities. The LC-MS analysis of the ethanol extract identified 14 compounds (Table 6).

Seventeen compounds were identified in the ethyl acetate fraction using LC-MS. Similar to the ethanol extract, nevadensin and chlorogenic acid were identified as the major compounds in this fraction, along with caffeic acid.

**Table 5. The IC$_{50}$ (µg/mL) for different extracts of C. montanus tested on three cell lines (A549, H1299, and Fibroblasts).**

| *C. montanus* extracts | A549 cancer cell (µg/mL) | H1299 cancer cell (µg/mL) | Fibroblast (µg/mL) |
|---|---|---|---|
| Ethyl acetate | 468.035 | 877.550 | Nontoxic |
| Ethanol | 476.545 | 641.175 | Nontoxic |
| Methanol | Nontoxic | Nontoxic | Nontoxic |
| Water | Nontoxic | 984.400 | Nontoxic |

Data are represented as mean ± SD, n=3.

### 3.6. Ethanol extract-loaded nanoemulsions

The influence of different concentrations on NE formulation was studied to identify the optimal concentrations of each component in the *V. montana* ethanol extract-loaded NE. In S1, the concentration of *V. montana* ethanol extract was 500 µg, combined with 1.75 mL of ethanol, 1.25 mL of Tween 20, 0.5 mL of oleic acid, and 1.0 mL of deionized water. After one week, S1 turned brown and showed notable aggregation, with MDS of 526.8 nm. The MDS of S1 NE was higher than the acceptable size range of nanoemulsions (typically <100 nm) [23]. Therefore, S1 was excluded from further studies. The NE S2 was prepared with 250 µg of *V. montana* ethanol extract, 2.25 mL of ethanol, 1.5 mL of Tween 20, 0.5 mL of oleic acid, and 0.75 mL of deionized water. The S2 appeared pale yellow and exhibited significant aggregation, indicating instability and prompting its exclusion from further analysis. S3 NE was formulated with 250 µg of *V. montana* ethanol extract, 2.25 mL of ethanol, 1.25 mL of Tween 20, 0.5 mL of oleic acid, and 1.00 mL of deionized water. The resulting MDS, PDI, and ZP measured 308.1 ± 21.1 nm, 0.2 ± 0.01, and −49.9 ± 1.5 mV, respectively. The MDS of S3 exceeded the acceptable nanoemulsion size range (<100 nm) [23]. Consequently, S3 was also excluded from further characterization. Additionally, S4 NE was prepared by reducing the volume of oleic acid to 0.25 mL while keeping the other components constant (250 µg of *V. montana* ethanol extract, 2.25 mL of ethanol, 1.25 mL of Tween 20, and 1.25 mL of deionized water). S4 NE showed initial measurements of MDS, PDI, and ZP being 13.6 ± 0.5 nm, 0.1 ± 0.05, and −1.7 ± 0.6 mV, respectively. Its droplet size was within the acceptable range for NE (below 100 nm) [23]. Therefore, S4 was selected for further characterization and subjected to thermodynamic and physical stability studies.

### 3.7. Characterization of ethanol extract-loaded nanoemulsion (NE)

**3.7.1. Thermodynamic stability.** Centrifugation: The stability of the S4 NE was evaluated by centrifugation at 3500 rpm for 15 min. post-centrifugation, S4 NE remained homogeneous and transparent, confirming its stability.

Heating/Cooling and Freezing/Thawing Cycles: To further assess stability, S4 - was subjected to six heating and cooling cycles and three freezing and thawing cycles. No signs of instability, such as cracking, creaming, or phase separation, were observed during these cycles, confirming the stability of the NE.

**3.7.2. pH Measurement.** The initial pH measurement of S4 was 5.39 ± 0.06. After 1 and 2 months of storage at RT (25 ± 2 °C), the pH was 5.36 ± 0.04 and 5.02 ± 0.15, respectively. It has been shown that the pH of liquid preparations intended for pulmonary delivery should ideally fall between 3.0 and 8.5, corresponding to the pH of lung fluid [24]. Hence, the pH values maintained within the acceptable range specified by the European Pharmacopeia for pulmonary medicines, making these formulations suitable for lung delivery [24].

**3.7.3. Viscosity measurement.** The viscosity curve of S4 NE demonstrated a shear-thinning behavior, as illustrated in Fig 2. This behavior indicates that the material becomes less viscous as the shear rate increases [25].

**3.7.4. Morphology and size of S4 using TEM.** The TEM images of S4 - revealed that the droplets were uniformly distributed, distinct, and spherical, with no aggregation. The droplet sizes measured by TEM were 12.49 nm and 20.14 nm, as shown in Fig 3, which closely aligned with the results obtained using a nanosizer of 13.6 ± 0.5 nm.

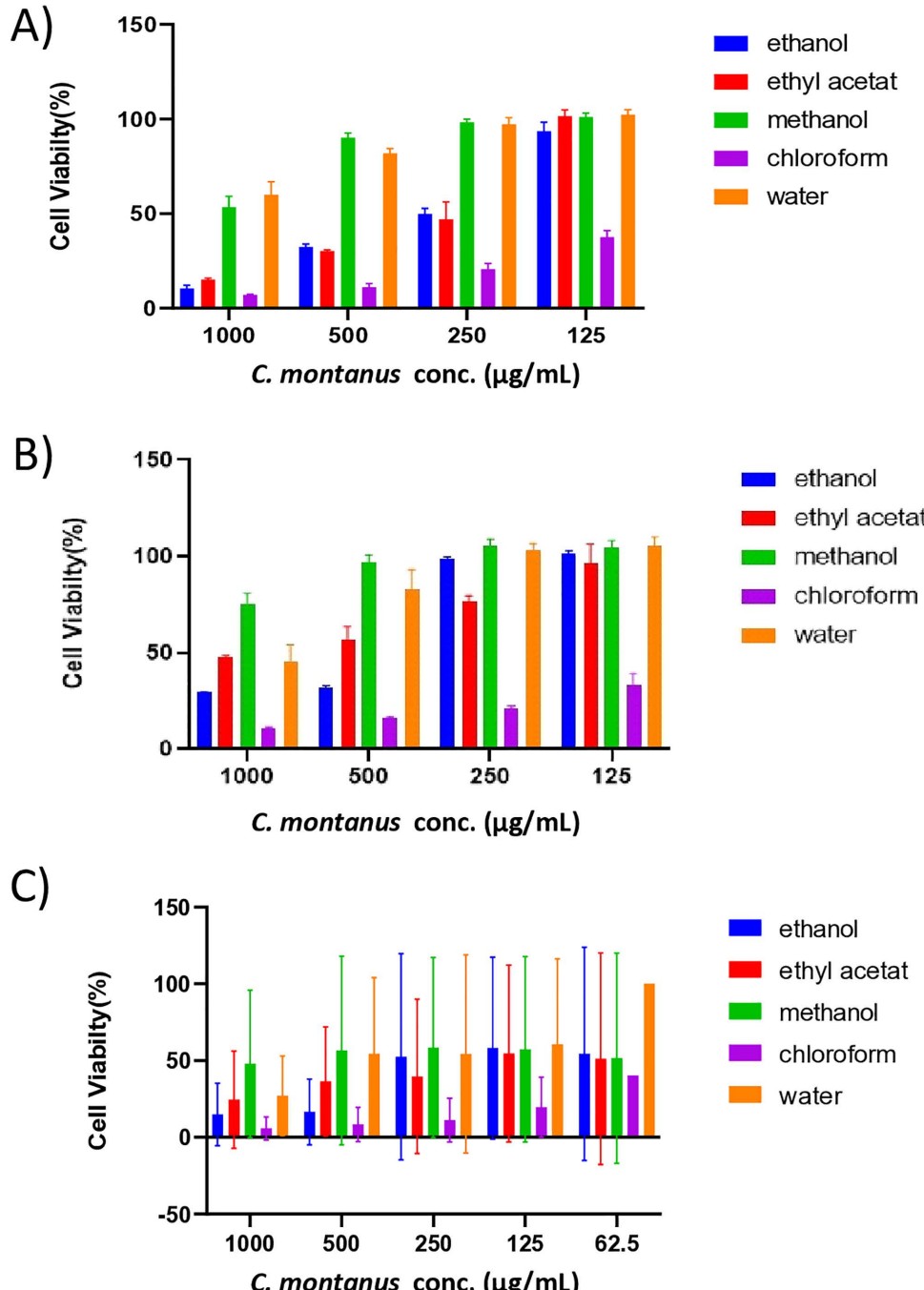

**Fig 1. Antiproliferative effect of C. montanus extracts on A) A549; B) H1299; and C) fibroblasts.** Data are represented as mean ± SEM, n = 3.

**3.7.5. Physical stability.** The physical evaluation of S4 NE after two months of storage at RT and 4 °C revealed that the formulation remained transparent and uniform with no color change or phase separation.

**3.7.6. Stability of mean droplet size (MDS), polydispersity index (PDI), and zeta potential (ZP) measurements.** The stability of S4 NE was evaluated by monitoring changes in MDS, PDI, and ZP over a two-month

Table 6. Identified phytochemicals in the ethanol and ethyl acetate extracts of C. montanus by LC-MS, expressed as % within identified phytochemicals.

| Rt | Phytochemicals | Molecular Formula | Ethanol extract | Ethyl-acetate extract |
|---|---|---|---|---|
| 1.9 | 2,5-dihydroxybenzoic acid | $C_7H_6O_4$ | – | 10.6 |
| 3.1 | Chlorogenic acid | $C_{16}H_{18}O_9$ | 22.5 | 16.7 |
| 3.5 | Caffeic acid | $C_9H_8O_4$ | 3.5 | 20.6 |
| 5.7 | Scopoletin | $C_{10}H_8O_4$ | 1.1 | 0.1 |
| 5.8 | Taxifolin | $C_{15}H_{12}O_7$ | – | 3.9 |
| 6.0 | Spiraeoside | $C_{21}H_{20}O_{12}$ | 8.9 | 8.9 |
| 6.4 | Aesculetin | $C_9H_7O_4$ | 1.3 | – |
| 6.2 | Rosmarinic acid | $C_{18}H_{16}O_8$ | 1.0 | 0.1 |
| 6.3 | 5-Glc tricin | $C_{23}H_{24}O_{12}$ | – | 3.2 |
| 6.7 | Quercetin | $C_{21}H_{20}O_{11}$ | – | 7.5 |
| 7.0 | Myricetin | $C_{15}H_{10}O_8$ | 0.9 | 0.1 |
| 7.3 | Aesculetin | $C_9H_6O_4$ | – | 0.1 |
| 8.9 | 3,6,2,3-tetrahydroxyflavone | $C_{15}H_{10}O_6$ | – | 2.4 |
| 9.3 | Quercetin-4′-O-malonylglucoside | $C_{24}H_{22}O_{15}$ | 1.5 | 0.3 |
| 9.5 | 6-methoxyluteolin | $C_{16}H_{12}O_7$ | 0.2 | 2.2 |
| 10.4 | Kaempferol | $C_{15}H_{10}O_6$ | 1.7 | – |
| 10.8 | Silymarin | $C_{25}H_{22}O_{10}$ | 0.5 | 0.1 |
| 11.0 | 5,6,4-trihydroxy-7.3-dimethoxy flavone | $C_{17}H_{14}O_7$ | 0.9 | 0.1 |
| 11.1 | Sorbifolin | $C_{16}H_{12}O_6$ | 14.4 | – |
| 13.3 | Nevadensin | $C_{18}H_{16}O_7$ | 41.6 | 23.1 |

period under two storage conditions: RT and 4 °C. Initial measurements showed that MDS and PDI values were within the acceptable range for NEs of <100 nm and < 0.5, respectively [23,26].

Over time, MDS, PDI, and ZP of S4 exhibited a gradual increase, nevertheless the MDS remained below 100 nm, indicating good stability [23]. Additionally, the PDI values were ≤0.5, indicating uniformity in droplet size distribution and affirming their homogeneity [26]. Furthermore, although ZP values were relatively low (i.e., less than −30 mV), the stability of S4 NE is attributed to the steric stabilization, resulting from the thick layer of the non-ionic surfactant Tween 20 with long side chains that surround the NE droplets and prevent coalescence [27]. Detailed numerical results for MDS, PDI, and ZP at each time point are provided in Table 7.

The stability studies confirmed that S4 NE maintained its physicochemical properties, including droplet size, PDI, and zeta potential, within acceptable ranges over two months of storage at both RT and 4 °C. The gradual increase in droplet size and PDI did not compromise the nanosized nature of the formulation, while the zeta potential measurements indicated sufficient stability despite being below the conventional threshold. These findings highlight the robustness of S4 NE for potential therapeutic applications [28,29].

### 3.8. Cytotoxicity evaluation of C. montanus NE

The cytotoxic effects of C. montanus NE were evaluated against A549 and H1299 cancer cell lines and normal fibroblasts using the MTT assay (Table 8 and Fig 4).

Experiments were performed in triplicate, and results are expressed as mean ± SD. A549 and H1299 cells exhibited similar sensitivity to the NE, while no cytotoxicity was observed against fibroblasts, indicating its selectivity for cancer cells and safety for normal cells.

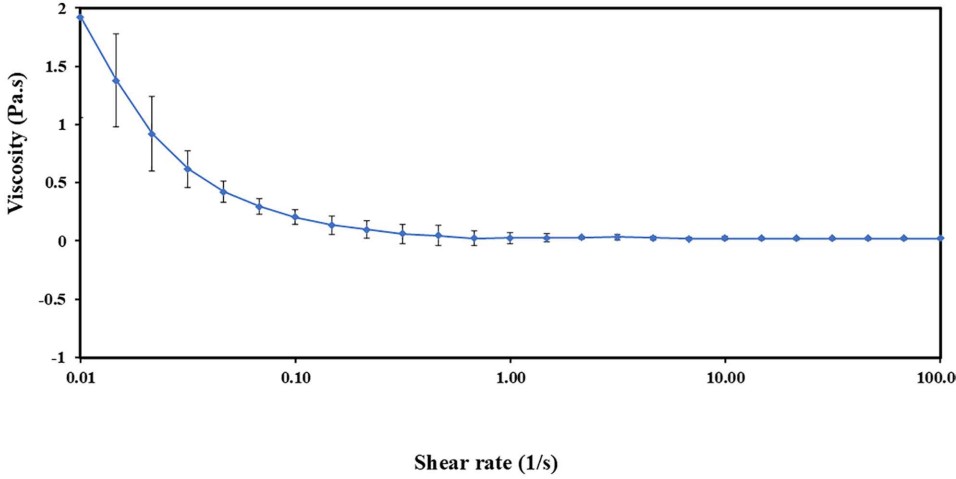

**Fig 2. Viscosity curve of S4 -.** Data is presented as mean ± SD (n = 3).

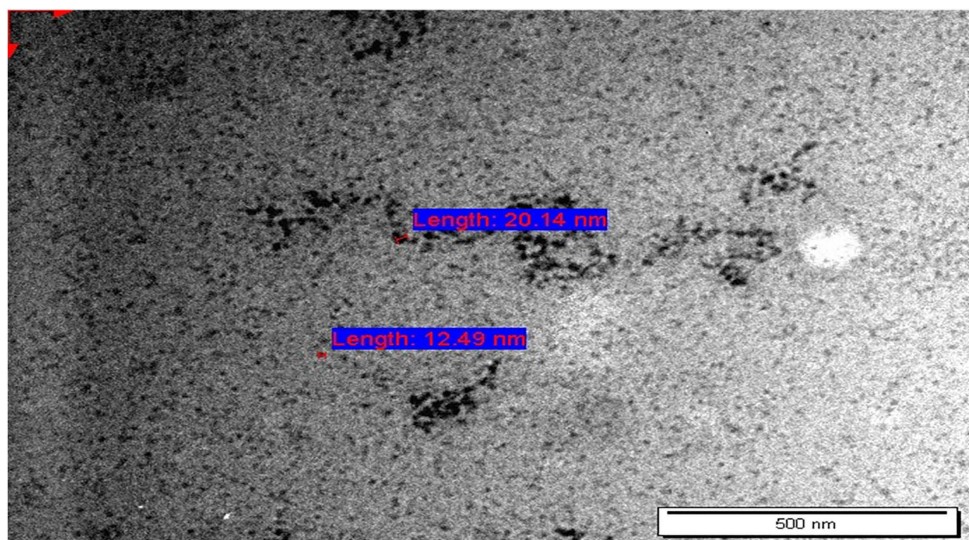

**Fig 3. TEM image of S4 - of spherical-shaped droplets.**

## 4. Discussion

The phytochemical analysis of C. montanus revealed a diverse composition of volatile and non-volatile bioactive compounds. A recent study by Abu-Darwish et al. (2022) identified sesquiterpene hydrocarbons (47.97%) and oxygenated sesquiterpenes (35.32%) as the major volatile compounds in the aroma profile of wild-growing C. montanus collected from Jordan [13]. Their analysis utilized dried plant material, whereas the present study employed fresh plant material for SPME analysis. These differences may be attributed to environmental factors, such as soil characteristics, temperature variations, and other climatic conditions, in addition to laboratory methodologies. Notably, Abu-Darwish collected plant

**Table 7. The stability measurements of MDS, PDI, and ZP of S4 NEs at RT and 4 °C.**

| Time | Stored at RT | | | Stored at 4 °C | | |
|---|---|---|---|---|---|---|
| | MDS | PDI | ZP | MDS | PDI | ZP |
| Initial | 13.6 ± 0.5 | 0.1 ± 0.1 | −1.7 ± 0.6 | 13.6 ± 0.5 | 0.1 ± 0.05 | −1.7 ± 0.6 |
| 1 week | 13.9 ± 0.1 | 0.2 ± 0.0 | −6.7 ± 1.7 | 13.4 ± 0.2 | 0.2 ± 0.01 | −3.5 ± 1.7 |
| 2 weeks | 16.3 ± 0.1 | 0.3 ± 0.0 | −6.1 ± 0.4 | 12.6 ± 0.6 | 0.1 ± 0.01 | −6.0 ± 0.4 |
| 3 weeks | 20.5 ± 0.3 | 0.3 ± 0.0 | −6.8 ± 0.4 | 23.9 ± 0.5 | 0.3 ± 0.01 | −10.2 ± 1.6 |
| 1 month | 25.7 ± 0.7 | 0.3 ± 0.0 | −6.6 ± 3.0 | 17.8 ± 0.2 | 0.2 ± 0.04 | −12.2 ± 0.0 |
| 2 months | 44.4 ± 0.9 | 0.5 ± 0.0 | −2.8 ± 0.9 | 19.4 ± 0.4 | 0.2 ± 0.02 | −8.1 ± 0.3 |

MSD: Mean droplet size, PDI: Polydispersity index, ZP: Zeta potential, NEs: Nanoemulsions, RT: Room temperature. Data are presented as mean ± SD (n = 3).

**Table 8. The IC$_{50}$ (μg/mL) for C. montanus NE tested on A549, H1299, and fibroblasts cells.**

| | A549 (μg/mL) | H1299 (μg/mL) | Fibroblast (μg/mL) |
|---|---|---|---|
| *C. montanus* extract | 383.5 ± 7.061**** | 343 ± 6.240**** | 1038.3 ± 62.110 |

Data are represented as mean ± SD, n = 3. Statistical analysis was conducted using two-way ANOVA followed by Dunnett's multiple comparisons test. ****$p < 0.0001$.

material from the Mediterranean biogeographic zone of Jordan, while in this study, mature plants were sourced from the Irano-Turanian biogeographic zone [13,30].

The findings highlight the impact of extraction techniques on the volatile profile of C. montanus. Hydrodistillation yielded a broader spectrum of volatile compounds, particularly sesquiterpenes, while SPME selectively enriched monoterpenes, predominantly oxygenated forms. The significant abundance of α-phellandrene (28.60%) and 1,8-cineole (23.30%) suggests their pharmacological relevance to the ethanol extract, given their solubility properties [31,32].

TPC and TFC analyses demonstrated the influence of solvent polarity on the extraction efficiency of bioactive compounds. Ethyl acetate, a semi-polar solvent, yielded the highest TPC (47.94 ± 0.32 mg GAE/g of DW) and TFC (20.34 ± 1.48 mg rutin/g of DW), followed by ethanol, which also exhibited considerable extraction efficiency (TPC: 20.54 ± 0.41 mg GAE/g of DW; TFC: 10.27 ± 0.74 mg rutin/g of DW). In contrast, water and non-polar solvents like petroleum ether showed significantly lower extraction capacities. The strong correlation between TPC and TFC suggests that the polyphenolic fraction of C. montanus is predominantly composed of flavonoids, which are known for their antioxidant and cytotoxic activities [33,34]. These results, in conjunction with antioxidant activity assessments, underscore the phytochemical richness of C. montanus and its potential as a therapeutic botanical resource, particularly when extracted using ethyl acetate and ethanol.

Cytotoxicity evaluations revealed that the ethanol extract and ethyl acetate fraction exhibited promising selective cytotoxic effects on cancer cell lines while sparing normal fibroblasts. Recent studies further support the pharmacological potential of C. montanus. Elhady et al. [8] reported that C. montanus extract and its isolated flavonoid, jaceidin, exhibited strong cytotoxic activity against HepG2 and MCF-7 cells, with in vivo studies demonstrating tumor-suppressing effects through VEGF inhibition and oxidative stress modulation. Additionally, Aidy et al. [9] found that the methanol and petroleum ether extracts of C. montanus displayed significant cytotoxicity against colorectal cancer cell lines and promoted the differentiation of colorectal cancer stem cell through the modulation of key genes and microRNA expressions.

The ethanol extract's major identified compound, nevadensin (5,7-dihydroxy-6,8,4′-trimethoxyflavone), is known for its diverse biological activities, including hypotensive, anti-inflammatory, antimicrobial, and anticancer properties [35]. Studies

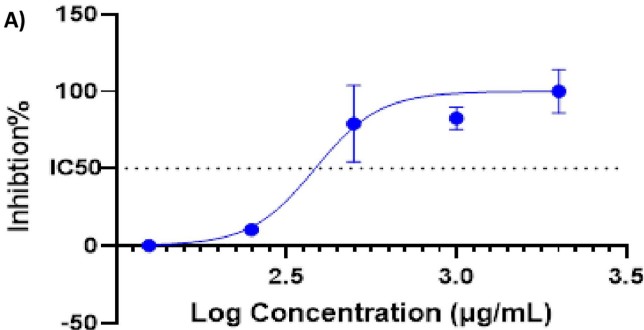

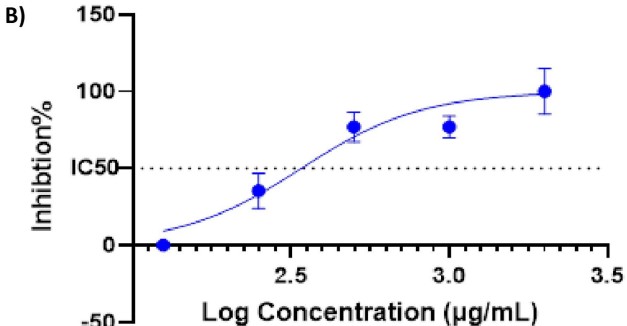

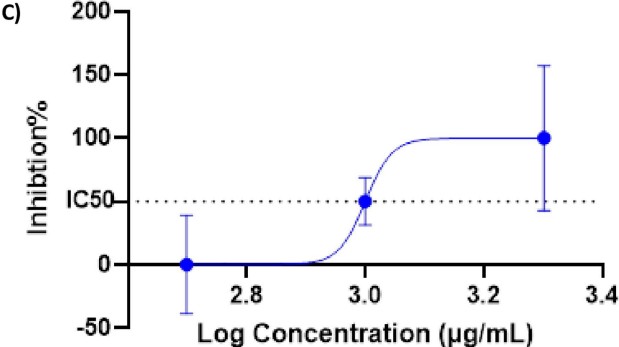

**Fig 4. Cytotoxicity profile and IC$_{50}$ of C. montanus NE when used to treat A) A549; B) H1299; and C) Fibroblasts cell lines.** The X-axis shows the log concentration in μM/mL, and the Y-axis shows the percentage of inhibited cells.

have identified nevadensin as a selective inhibitor of human carboxylesterase 1 and topoisomerases TOPO I and TOPO IIα using in vitro models [36,37].

Chlorogenic acid, a widely distributed phenolic acid, exhibits multiple biological effects, including hepatoprotective, antimicrobial, immunomodulatory, antioxidant, antidiabetic, and anticancer properties. Studies have demonstrated its role in cancer prevention through mechanisms such as inhibition of cell division, suppression of angiogenesis, cell cycle arrest, and apoptosis induction, supported by in vitro, in vivo, and clinical studies [38,39]. Notably, chlorogenic acid has been approved by the China Food and Drug Administration as a natural anticancer drug [38].

Sorbifolin (Scutellarin-7-methylether), another major compound in the ethanol extract, is recognized for its antioxidant, anticancer, antimicrobial, anti-inflammatory, and antidiabetic activities [40]. In silico studies by Shah and Kin concluded that sorbifolin is a potent candidate for Ewing sarcoma treatment [41].

These findings suggested that the identified phytochemicals, particularly flavonoids, play a pivotal role in the cytotoxic activity observed with the ethanol extract. Flavonoids and phenolic compounds are well-recognized for their strong anti-oxidant capacity, primarily mediated through redox mechanisms that enable them to scavenge reactive oxygen species (ROS) and modulate oxidative stress pathways. This radical-neutralizing ability not only contributes to cellular protection but may also underlie the observed antiproliferative effects by triggering apoptosis, cell cycle arrest, or interference with signaling cascades involved in tumor progression [42]. Such dual antioxidant–antiproliferative properties highlight their therapeutic relevance. Furthermore, recent investigations into in vitro callus induction have gained increasing attention, as this biotechnological approach can stimulate the accumulation of secondary metabolites, including flavonoids and pheno-lics, in controlled culture systems. Enhanced metabolite production through callus cultures may potentiate the biological efficacy of plant extracts, offering a sustainable and reproducible source of bioactive compounds for pharmacological applications. [43,44].

To enhance pulmonary delivery for lung carcinoma treatment, nanoemulsions (NEs) were developed. The optimized S4 NE formulation demonstrated excellent stability and an appropriate droplet size for pulmonary administration. Cytotoxicity evaluations revealed that C. montanus NE selectively targeted cancer cells (A549 and H1299) while sparing normal fibro-blasts, supporting its potential as a safe and effective anticancer formulation. These findings emphasize the therapeutic promise of C. montanus NE for targeted cancer therapy, warranting further investigation into its mechanisms of action and in vivo efficacy.

## 5. Conclusions

This comprehensive study on *C. montanus* highlights its significant phytochemical diversity and pharmacological potential. The GC-MS analysis revealed that HD, produces a more diverse composition rich in sesquiterpenes, while SPME was selectively enriched with monoterpenes. The quantitative analysis of TPC and TFC demonstrated that ethyl acetate frac-tion and ethanol extract contain large amounts of these bioactive compounds, aligning with their strong antioxidant and cytotoxic activities. The ethanol extract and ethyl acetate fraction showed promising selective cytotoxicity against A549 and H1299 cancer cell lines, without affecting normal fibroblasts, underlining their potential for therapeutic applications.

LC-MS analysis identified key phytochemicals, including nevadensin and chlorogenic acid, which likely contribute to the observed bioactivities. The formulation of ethanol extract-loaded NEs further enhanced the delivery of *C. montanus* extracts, with S4 NE demonstrating excellent stability, with suitable droplet size, and suitable pH for pulmonary delivery. The cytotoxicity evaluation of the NE confirmed its selective anticancer activity, with no adverse effects on normal cells.

Overall, this study underscores the therapeutic potential of *C. montanus* as a rich source of bioactive compounds with antioxidant, cytotoxic, and anticancer properties. The findings suggest that *C. montanus* extracts, particularly when for-mulated as NEs, are promising natural therapeutic agents, especially in cancer treatment. Future studies should focus on elucidating the mechanisms of action of the identified compounds and exploring their *in vivo* efficacy and safety profiles.

## Author contributions

**Conceptualization:** Nour Aboalhaija, Fatma Afifi.

**Data curation:** Nour Aboalhaija.

**Formal analysis:** Nour Aboalhaija.

**Funding acquisition:** Nour Aboalhaija.

**Investigation:** Nour Aboalhaija, Fatma Afifi.

**Methodology:** Nour Aboalhaija, Hala Abulawi, Heba Syaj, Ismail Abaza.

**Project administration:** Nour Aboalhaija.

**Resources:** Nour Aboalhaija.

**Software:** Nour Aboalhaija, Mohammad Alwahsh, Elham Abusharieh.

**Supervision:** Nour Aboalhaija, Fatma Afifi.

**Validation:** Nour Aboalhaija, Rania Hamed, Mohammad Alwahsh.

**Visualization:** Nour Aboalhaija.

**Writing – original draft:** Hala Abulawi, Heba Syaj.

**Writing – review & editing:** Nour Aboalhaija, Rania Hamed, Mohammad Alwahsh, Fatma Afifi.

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
