## [Editor Report · Decision Letter 0]

29 Apr 2025

Dear Dr. Aboalhaija,

Thank you for submitting your manuscript to PLOS ONE. After careful consideration, we feel that it has merit but does not fully meet PLOS ONE’s publication criteria as it currently stands. Therefore, we invite you to submit a revised version of the manuscript that addresses the points raised during the review process.

**Please update the Latin name of the study species. *Varhemia montana * is actually a synonym of *Chiliadenus montanus* (Vahl) Brullo (e.g., **
https://powo.science.kew.org/taxon/urn:lsid:ipni.org:names:257204-1). **The whole first paragraph of the introductory section needs thorough revision, as the taxonomy of the former genus *Varhemia * has been changed since 2018. Please add line and page numbers to facilitate the future review process. Please explain letters in superscripts in Table 1. **

We look forward to receiving your revised manuscript.

Kind regards,

Branislav T. Šiler, Ph.D.

Academic Editor

PLOS ONE

“This research was funded by Deanship of Scientific Research and Innovation at Al-Zaytoonah University of Jordan (2022-2023/07/23).”

---

## [Author Response · Author response to Decision Letter 1]

9 May 2025

Re: Manuscript ID: PONE-D-25-17902

Dear Dr. Šiler,

We sincerely thank you and the reviewers for your constructive and insightful comments on our manuscript. We appreciate the opportunity to revise and resubmit our work.

We have carefully considered each comment and revised the manuscript accordingly. In brief:

1. Please update the Latin name of the study species. Varhemia montana is actually a synonym of Chiliadenus montanus (Vahl) Brullo (e.g., https://powo.science.kew.org/taxon/urn:lsid:ipni.org:names:257204-1).

The Latin name of the plant species has been updated to Chiliadenus montanus (Vahl) Brullo, in accordance with current taxonomic classifications.

2. The whole first paragraph of the introductory section needs thorough revision, as the taxonomy of the former genus Varhemia has been changed since 2018.

The first paragraph of the introduction has been thoroughly revised to reflect this change.

3. Please add line and page numbers to facilitate the future review process.

Line and page numbers have been added throughout the manuscript to aid future review.

4. Please explain letters in superscripts in Table 2.

The superscript letters in Table 2 have been clearly explained in a new footnote to enhance clarity.

5. Figure Reorganization: As per PLOS ONE’s figure formatting guidelines.

We have uploaded updated versions of all figures. Figure 1a–c has been split into three separate figures to meet the requirements. We have also split Figure 4 as necessary and uploaded the revised versions accordingly.

We hope the revised version meets your expectations and the requirements of PLOS ONE. Thank you again for your time and consideration.

Nour Aboalhaija, PhD

Assistant Professor of Phytochemistry

Department of Pharmacy

Faculty of Pharmacy

Al-Zaytoonah University of Jordan

Mobile: +962-79-6304081

E-mail: n.aboalhaija@zuj.edu.jo

Google Scholar: https://scholar.google.com/citations?user=CBVZdtcAAAAJ&hl=en

ResearchGate: https://www.researchgate.net/profile/Nour-Aboalhaija

---

## [Decision Letter · Decision Letter 1]

3 Sep 2025

Dear Dr. Aboalhaija,

Thank you for submitting your manuscript to PLOS ONE. After careful consideration, we feel that it has merit but does not fully meet PLOS ONE’s publication criteria as it currently stands. Therefore, we invite you to submit a revised version of the manuscript that addresses the points raised during the review process.

The authors need to provide additional clarification of the methodology used and to add the missing results on antioxidant activity. The Discussion section should be thoroughly revised by comparing the obtained results with those published earlier, as suggested by Reviewer #2. Regarding the Reviewer #2's comment on the capitalization of compound names, please keep them non-capitalized in the text and capitalized in Table 2. Please provide thorough feedback with point-by-point response to reviewers' comments. Note that PLOS ONE do not copyedit the text before publishing. Therefore, I suggest the authors having it thoroughly checked by a native English speaker or a professional editing agency to polish the language. Please be aware the additional reviewers' comments provided in the attached files. 

We look forward to receiving your revised manuscript.

Kind regards,

Branislav T. Šiler, Ph.D.

Academic Editor

PLOS ONE

Journal Requirements:

Reviewers' comments:

Reviewer's Responses to Questions

**Comments to the Author**

Reviewer #1: (No Response)

Reviewer #2: (No Response)

2. Is the manuscript technically sound, and do the data support the conclusions?

Reviewer #1: Yes

Reviewer #2: Yes

3. Has the statistical analysis been performed appropriately and rigorously?

Reviewer #1: Yes

Reviewer #2: I Don't Know

4. Have the authors made all data underlying the findings in their manuscript fully available?

Reviewer #1: Yes

Reviewer #2: Yes

5. Is the manuscript presented in an intelligible fashion and written in standard English?

Reviewer #1: Yes

Reviewer #2: Yes

Reviewer #1: i would like to acknowledge the effort of the authors in this paper, in order to improve the quality of this paper we recommand the application of the following remarks:

1. Please bring back the 'Amman 11733, Jordan' to the upper line.

2. correct the abstract.

3. please check PLOS One guide for authors. the citations have to be between brackets [1], check the following link: https://journals.plos.org/plosone/s/submission-guidelines#loc-references”

4. the GC-MS method, some parameters are missing please add them they are as follow: the apparatus, the column, the oven temperature, the detector type and temperature, the vector gas, the flow rate...

5. please add the equation using the microsoft word equation insert; add number of equation (1), and cite it on the text (using the formula (1) below).

6. add reference of this method

7. for each result please add how is it calculated (an equation or extrapolation from graph.

8. the LC-MS analysis part is not complete, please add the following informations, which solvent is used to prepare the samples, add the type of column you used as well as its dimentions, you also must add the temperatures, the flowrate and the the elution system (whhich solvents, gradient or isocratic and the ratios), which system have you used for the injection, have you used internal standards ?

9. down the tables add the abreviations.

10. more details about monitoring the physical stability, is it with an apparatus, how is it quantified, and what is the principle of this method

11. for all those calculations add the informations related to, which apparatus was used, how is the value calculated (equation, equation extrapolation or given by the apparatus), add also the unit of measurement.

12. in table 3, please center the cells content (Extract and TFC are up, when TPC is centered)

13. in table 6, add MDS, PDI, ZP as abreviation down the table

Please check carefully the comments in the pdf file, Thank you

Reviewer #2: This study addresses an interesting and timely topic. The obtained results are significant and may benefit future research. However, before publishing this work, it is necessary to make certain corrections and clarify some doubts. I hope that my comments will contribute to improving the quality of the manuscript.

**Do you want your identity to be public for this peer review?** For information about this choice, including consent withdrawal, please see our Privacy Policy

Reviewer #1: **Yes: ** Hicham Mechqoq

Reviewer #2: No

---

## [Author Response · Author response to Decision Letter 2]

12 Sep 2025

Reviewer #1: i would like to acknowledge the effort of the authors in this paper, in order to improve the quality of this paper we recommand the application of the following remarks:

1. Please bring back the 'Amman 11733, Jordan' to the upper line.

Response: The address has been corrected as requested

2. correct the abstract.

Response: The abstract has been corrected as requested.

3. please check PLOS One guide for authors. the citations have to be between brackets [1], check the following link: https://journals.plos.org/plosone/s/submission-guidelines#loc-references”

Response: All references have been reformatted to the required style in accordance with PLOS ONE guidelines.

4. the GC-MS method, some parameters are missing please add them they are as follow: the apparatus, the column, the oven temperature, the detector type and temperature, the vector gas, the flow rate...

Response: Full GC-MS details (apparatus, column, oven temperature program, detector type and temperature, carrier gas, and flow rate) have been added to the Materials and Methods section (2.3. Gas Chromatography-Mass Spectrometry).

5. please add the equation using the microsoft word equation insert; add number of equation (1), and cite it on the text (using the formula (1) below).

Response: The equations 1 and 2 have been inserted using the MS Word equation editor, numbered accordingly and cited within the text.

6. add reference of this method

Response: Relevant references have been included.

7. for each result please add how is it calculated (an equation or extrapolation from graph.

Response: For each reported result, we have clarified how values were obtained, whether calculated by equation, extrapolated from graphs, or directly obtained from instruments.

8. the LC-MS analysis part is not complete, please add the following informations, which solvent is used to prepare the samples, add the type of column you used as well as its dimentions, you also must add the temperatures, the flowrate and the the elution system (whhich solvents, gradient or isocratic and the ratios), which system have you used for the injection, have you used internal standards ?

Response: All the requested details regarding LC-MS have been included in the methods section (2.8. LC-MS Analysis).

9. down the tables add the abreviations.

Response: Abbreviations have been added below each table where relevant

10. more details about monitoring the physical stability, is it with an apparatus, how is it quantified, and what is the principle of this method.

Response: The physical stability monitoring method has been fully described, including apparatus, principles of measurement, and parameters observed

11. for all those calculations add the informations related to, which apparatus was used, how is the value calculated (equation, equation extrapolation or given by the apparatus), add also the unit of measurement.

Response: Additional details regarding apparatus, calculation methods, and units have been added for clarity and reproducibility.

12. in table 3, please center the cells content (Extract and TFC are up, when TPC is centered)

Response: The alignment of cells in Table 3 has been corrected

13. in table 6, add MDS, PDI, ZP as abreviation down the table

Response: Abbreviations (MDS, PDI, ZP) have been added below Table 7 in the new version.

Please check carefully the comments in the pdf file, Thank you

Response: We have carefully reviewed and addressed all in-file comments and annotations in the attached PDF. The corresponding revisions have been implemented throughout the manuscript and summarized point-by-point in this letter. All edits are visible in the Revised Manuscript with Track Changes. This includes corrections to figure labels, table abbreviations, terminology consistency, and alignment of numerical values with the tables. The only exception is the comment regarding adding a reference next to each activity. Since several references were cited multiple times for different activities, we opted to avoid unnecessary repetition and instead maintained a clear, consolidated referencing style.

Reviewer #2: This study addresses an interesting and timely topic. The obtained results are significant and may benefit future research. However, before publishing this work, it is necessary to make certain corrections and clarify some doubts. I hope that my comments will contribute to improving the quality of the manuscript.

In the Abstract, you say: „Spontaneous emulsification was used to formulate ethanolic NEs, with S3, S4, S5, and S6 demonstrating thermodynamic stability, suitable pulmonary pH, and droplet sizes below 200 nm.”… but in the Material and Method section, you indicate that further analyses of thermodynamic stability were done on S4. Please additionally explain what was done and for which NE.

Response: We thank the reviewer for pointing that out. The abstract, Materials and methods section, and Results section have been revised to describe the preparation of ethanol extract-loaded nanoemulsions (S1-S4 NEs) and identify the optimal S4 NE that underwent further analyses.

Spontaneous emulsification was used to formulate ethanolic nanoemulsions (S1-S4 NEs) with varying concentrations of ethanol extract, surfactant, cosurfactant, and oil phase. The optimal S4 NE demonstrated thermodynamic stability, appropriate pulmonary pH, and droplet sizes below 100 nm.

New section has been added (3.6. Ethanol Extract-loaded Nanoemulsions)

Page 12, line 127: Indicate which cancer cell lines (lung cancer) were used in the study.

Response: The cancer cell lines used in this study have been clearly specified in the revised Methods section; Human lung cancer cell lines A549, H1299.

Page 12, line 136: Add the used concentration of MT reagent.

Response: The concentration of the MTT reagent has been added to the Methods section; 15 μL (0.5mg/mL in each well) of MTT.

Page 13, line 127: Add an explanation for S4 selection for thermodynamic studies.

Response: We thank the reviewer for pointing that out. The following paragraph has been added to explain the selection of S4 NE for thermodynamic stability.

S4 NE was prepared by reducing the volume of oleic acid to 0.25 mL while keeping the other components constant (250 µg of V. montana ethanol extract, 2.25 mL of ethanol, 1.25 mL of Tween 20, and 1.25 mL of deionized water). S4 NE showed initial measurements of MDS, PDI, and ZP being 13.6 ± 0.5 nm, 0.1 ± 0.05, and -1.7 ± 0.6 mV, respectively. Its droplet size was within the acceptable range for NE (below 100 nm) [1]. Therefore, S4 NE was selected for further characterization and subjected to thermodynamic and physical stability studies.

This explanation has been added to the revised version of the manuscript.

Page 13, line 165: Specify which physical characteristics are monitored.

Response: We thank the reviewer for pointing that out. The physical characteristics, including phase separation, homogeneity, and color change, of the optimal S4 NE have been monitored visually and added to the revised manuscript. The following sections has been modified as follows:

2.10.5. Physical Stability

Physical stability of C. montanus NEs, including phase separation, homogeneity, and color change, was monitored visually for 60 days at RT and at 4 °C.

3.7.5. Physical Stability

The physical evaluation of S4 NE after two months of storage at RT and 4 °C revealed that the nanoemulsion remained transparent and uniform with no color change or phase separation.

In the Materials and Methods, rotate the subheadings 2.11.2 and 2.11.3 so that it is displayed the same as in the Results (or do the rotation in the Results).

Response: We thank the reviewer for pointing that out. The subheadings 2.11.2 and 2.11.3 have been rotated in the Methods and Results Sections.

Page 14, line 185: „hydrocarbon monoterpenes (37.7%) and oxygenated monoterpenes (36.9%) as shown in Table 2.”… There is a mistake here or in the table regarding the values.

Response: Consistencies between text and Table 2 values have been corrected.

Page 14, line 193: “1,8-cineole (35.60%), sabinene (12.70%), and α-artemisia triene (11.40%). “…Is it necessary to capitalise compound names as written in Table 2? If so, implement the changes throughout the entire manuscript.

Response: Compound names are lowercase in the text and capitalized in Table 2, following journal guidelines.

Same line: “1,8-cineole (35.60%)” …value does not correspond to value from Table 2.

Response: Consistencies between text and Table 2 values have been corrected.

In the text of the manuscript, you use the term aqueous extract (Page 16, line 202), but in the tables and figures, you use the term water extract. Use one term and make the appropriate changes. Also, the order in which extracts are displayed is not uniform in tables and graphs. Please correct it. Apply the selected term and order of extracts to the figures, also.

Response: The terminology has been standardized across the text, tables, and figures, using “water extract.” The order of extracts has also been made uniform.

Results for Antioxidant activity are not displayed.

Response: We respectfully note that the results on antioxidant activity were originally presented in Section 3.3 (Antioxidant Activity) in textual form. In the revised version, these results have been consolidated into new table (Table 4) and are further elaborated upon in both the Results and Discussion sections.

The plant name remained unchanged on the figures.

Response: The plant name has been corrected and updated consistently across all figures.

The Discussion section should be expanded by comparing the obtained results with the literature data, as well as emphasising the significance of the results of this work. For example, the results of NE cytotoxicity indicate a significantly reduced IC50 value of the used NE compared to the corresponding extract, but this is not emphasised or interpreted anywhere in the work.

Response: We thank the reviewer for pointing that out. The Discussion section has been expanded by comparing the obtained results with the literature data.

Recent investigations are focused on In vitro callus induction, which generally allows for enhanced production of certain plant extract components, thereby contributing to their enhanced efficacy. It would be useful to mention this in combination with the spontaneous emulsification of the extract as a suggestion for future research. Here are some proposed references:

Abu-Darwish, D., Shibli, R., & Al-Abdallat, A. M. (2022). In vitro cultures and volatile organic compound production in chiliadenus montanus (vhal.) brullo. Plants, 11(10), 1326.

Abu-Darwish, D., Shibli, R., & Al-Abdallat, A. M. (2024). Phenolic Compounds and Antioxidant Activity of Chiliadenus montanus (Vhal.) Brullo. grown in vitro. Jordan Journal of Pharmaceutical Sciences, 17(3), 611-628.

Response: A paragraph has been added to the Discussion suggesting in vitro callus induction and spontaneous emulsification of extracts as potential future research directions, and the recommended references have been cited (Abu-Darwish et al., 2022; Abu-Darwish et al., 2024).

---

## [Editor Report · Decision Letter 2]

16 Sep 2025

Phytochemical Characterization, Anticancer Potential, and Nanoemulsion-Based Delivery of Chiliadenus montanus

PONE-D-25-17902R2

Dear Dr. Aboalhaija,

We’re pleased to inform you that your manuscript has been judged scientifically suitable for publication and will be formally accepted for publication once it meets all outstanding technical requirements.

Kind regards,

Branislav T. Šiler, Ph.D.

Academic Editor

PLOS ONE
---

## [Editor Report · Acceptance letter]

PONE-D-25-17902R2

PLOS ONE

Dear Dr. Aboalhaija,

I'm pleased to inform you that your manuscript has been deemed suitable for publication in PLOS ONE. Congratulations! Your manuscript is now being handed over to our production team.

Kind regards,

on behalf of

Dr. Branislav T. Šiler

Academic Editor

PLOS ONE